# Self-supervised Object-Centric Learning for Videos

**Görkay Aydemir**[1]    **Weidi Xie**[3,4]    **Fatma Güney**[1,2]

[1] Department of Computer Engineering, Koç University    [2] KUIS AI Center
[3] CMIC, Shanghai Jiao Tong University    [4] Shanghai AI Laboratory
{gaydemir23, fguney}@ku.edu.tr   weidi@sjtu.edu.cn

## Abstract

Unsupervised multi-object segmentation has shown impressive results on images by utilizing powerful semantics learned from self-supervised pretraining. An additional modality such as depth or motion is often used to facilitate the segmentation in video sequences. However, the performance improvements observed in synthetic sequences, which rely on the robustness of an additional cue, do not translate to more challenging real-world scenarios. In this paper, we propose the first fully unsupervised method for segmenting multiple objects in real-world sequences. Our object-centric learning framework spatially binds objects to slots on each frame and then relates these slots across frames. From these temporally-aware slots, the training objective is to reconstruct the middle frame in a high-level semantic feature space. We propose a masking strategy by dropping a significant portion of tokens in the feature space for efficiency and regularization. Additionally, we address over-clustering by merging slots based on similarity. Our method can successfully segment multiple instances of complex and high-variety classes in YouTube videos.[1]

## 1  Introduction

Given a video sequence of a complex scene, our goal is to train a vision system that can discover, track, and segment objects, in a way that abstracts the visual information from millions of pixels into semantic components, *i.e.*, objects. This is commonly referred to as object-centric visual representation learning in the literature. By learning such abstractions of the visual scene, the resulting object-centric representation acts as fundamental building blocks that can be processed independently and recombined, thus improving the model's generalization and supporting high-level cognitive vision tasks such as reasoning, control, *etc*. [13, 29].

The field of object-centric representation learning in computer vision has made significant progress over the years, starting from synthetic images [34, 37], and has since shifted towards in-the-wild image [20, 49] and real-world videos [65, 85, 87, 46, 60]. In general, existing approaches typically follow an autoencoder training paradigm [52, 26], *i.e.*, reconstructing the input signals with certain bottlenecks, with the hope to group the regional pixels into semantically meaningful objects based on the priors of architecture or data. In particular, for images, low-level features like color, and semantic features from pretrained deep networks, are often used to indicate the assignment of pixels to objects, while for videos, additional modalities/signals are normally incorporated, such as optical flow [40], depth map [17], with segmentation masks directly available from the discontinuities. Despite being promising, using additional signals in videos naturally incurs computation overhead and error accumulation, for example, optical flow may struggle with static or deformable objects and large displacements between frames, while depth may not be readily available in generic videos and its estimation can suffer in low-light or low-contrast environments.

---

[1]Project page: https://kuis-ai.github.io/solv

37th Conference on Neural Information Processing Systems (NeurIPS 2023).

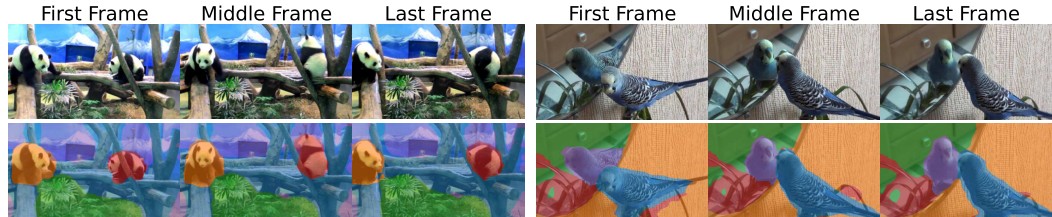

Figure 1: We introduce SOLV, an object-centric framework for instance segmentation in real-world videos. This figure depicts instance segmentation results of first, middle, and last frames of videos on Youtube-VIS 2019 [87]. Without any supervision both in training and inference, we manage to segment object instances accurately.

In this paper, we introduce SOLV, a self-supervised model capable of discovering multiple objects in real-world video sequences without using additional modalities [2, 36, 40, 17] or any kind of weak supervision such as first frame initialization [40, 17]. To achieve this, we adopt axial spatial-temporal slot attentions, that first groups spatial regions within frame, then followed by enriching the slot representations using additional cues from interactions with neighboring frames. We employ masked autoencoder (MAE)-type training, with the objective of reconstructing dense visual features from masked inputs. This approach has two benefits: first, it acts as an information bottleneck, forcing the model to learn the high-level semantic structures, given only partial observations; second, it alleviates memory constraints, which helps computational efficiency. Additionally, due to the complexity of visual scenes, a fixed number of slots often leads to an over-segmentation issue. We address this issue by merging similar slots with a simple clustering algorithm. Experimentally, our method significantly advances the state-of-the-art without using information from additional modalities on a commonly used synthetic video dataset, MOVi-E [25] and a subset of challenging Youtube-VIS 2019 [87] with in-the-wild videos (Fig. 1).

To summarize, we make the following contributions: (i) We propose a self-supervised multi-object segmentation model on real-world videos, that uses axial spatial-temporal slots attention, effectively grouping visual regions with similar property, without relying on any additional signal; (ii) We present an object-centric learning approach based on masked feature reconstruction, and slot merging; (iii) Our model achieves state-of-the-art results on the MOVi-E and Youtube-VIS 2019 datasets, and competitive performance on DAVIS2017.

## 2 Related Work

**Object-centric Learning:** The field of unsupervised learning of object-centric representations from images and videos has gained considerable attention in recent years. Several approaches have been proposed to address this problem with contrastive learning [31, 41, 84] or more recently with reconstruction objectives. An effective reconstruction-based approach first divides the input into a set of region identifier variables in the latent space, *i.e.* slots, that bind to distinctive parts corresponding to objects. Slot attention has been applied to both images [26, 5, 10, 12, 18, 50, 89, 19, 27, 15, 52, 69] and videos [35, 71, 28, 40, 33, 75, 90, 43]. However, these methods are typically evaluated on synthetic data and struggle to generalize to real-world scenarios due to increasing complexity. To address this challenge, previous works explore additional information based on the 3D structure [8, 58, 32] or reconstruct different modalities such as flow [40] or depth [17]. Despite these efforts, accurately identifying objects in complex visual scenes without explicit guidance remains an open challenge. The existing work relies on guided initialization from a motion segmentation mask [1, 2] or initial object locations [40, 17]. To overcome this limitation, DINOSAUR [67] performs reconstruction in the feature space by leveraging the inductive biases learned by recent self-supervised models [7]. We also follow this strategy which has proven highly effective in learning object-centric representations in real-world data without any guided initialization or explicit supervision.

**Object Localization from DINO Features:** The capabilities of Vision Transformers (ViT) [14] have been comprehensively investigated, leading to remarkable findings when combined with self-supervised features from DINO [7]. By grouping these features with a traditional graph partitioning method [57, 68, 81], impressive results can be achieved compared to earlier approaches [76, 77].

Recent work, CutLER [80], extends this approach to segment multiple objects with series of normalized cuts. The performance of these methods without any additional training shows the power of self-supervised DINO features for segmentation and motivates us to build on it in this work.

**Video Object Segmentation:** Video object segmentation (VOS) [86, 4, 22, 23, 42, 38, 44, 45, 56, 59, 61, 72, 78, 79, 88] aims to identify the most salient object in a video without relying on annotations during evaluation in unsupervised setting [73, 21, 47, 53] and only annotation of the first frame in semi-supervised setting [6]. Even if the inference is unsupervised, ground-truth annotations can be used during training in VOS [11, 16, 63, 54, 6]. Relying on labelled data during training might introduce a bias towards the labelled set of classes that is available during training. In this paper, we follow a completely unsupervised approach without using any annotations during training or testing.

Motion information is commonly used in unsupervised VOS to match object regions across time [36, 9, 51, 66]. Motion cues particularly come in handy when separating multiple instances of objects in object-centric approaches. Motion grouping [86] learns an object-centric representation to segment moving objects by grouping patterns in flow. Recent work resorts to sequential models while incorporating additional information [70, 40, 17, 1]. In this work, we learn temporal slot representations from multiple frames but we do not use any explicit motion information. This way, we can avoid the degradation in performance when flow cannot be estimated reliably.

# 3 Methodology

In this section, we start by introducing the considered problem scenario, then we detail the proposed object-centric architecture, that is trained with self-supervised learning.

## 3.1 Problem Scenario

Given an RGB video clip as input, $i,.e.$, $\mathcal{V}_t = \left\{ \mathbf{v}_{t-n}, \ldots, \mathbf{v}_t, \ldots \mathbf{v}_{t+n} \right\} \in \mathbb{R}^{(2n+1) \times H \times W \times 3}$, our goal is to train an object-centric model that processes the clip, and outputs all object instances in the form of segmentation masks, *i.e.*, discover and track the instances in the video, via *self-supervised learning*, we can formulate the problem as :

$$\mathbf{m}_t = \Phi \left( \mathcal{V}_t; \Theta \right) = \Phi_{\text{vis-dec}} \circ \Phi_{\text{st-bind}} \circ \Phi_{\text{vis-enc}} \left( \mathcal{V}_t \right) \tag{1}$$

where $\mathbf{m}_t \in \mathbb{R}^{K_t \times H \times W}$ refers to the output segmentation mask for the middle frame with $K_t$ discovered objects. After segmenting each frame, we perform Hungarian matching to track the objects across frames in the video. $\Phi \left( \cdot; \Theta \right)$ refers to the proposed segmentation model that will be detailed in the following section. Specifically, it consists of three core components (see Fig. 2), namely, visual encoder that extracts frame-wise visual features (Section 3.2.1), spatial-temporal axial binding that first groups pixels into slots within frames, then followed by joining the slots across temporal frames (Section 3.2.2), and visual decoder that decodes the spatial-temporal slots to reconstruct the dense visual features, with the segmentation masks of objects as by-products (Section 3.2.3).

## 3.2 Architecture

Generally speaking, our proposed architecture is a variant of Transformer, that can be trained with simple masked autoencoder, *i.e.*, reconstructing the full signals given partial input observation. However, unlike standard MAE [30] that recovers the image in pixel space, we adopt an information bottleneck design, that first assigns spatial-temporal features into slots, then reconstructs the dense visual features from latent slots. As a result, each slot is attached to one semantically meaningful object, and the segmentation masks can be obtained as by-products from reconstruction, *i.e.*, without relying on manual annotations.

### 3.2.1 Visual Encoder

Given the RGB video clip, we divide each image into regular non-overlapping patches. Following the notation introduced in Section 3.1, *i.e.*, $\mathcal{V}_t = \left\{ \mathbf{v}_{t-n}, \ldots, \mathbf{v}_t, \ldots \mathbf{v}_{t+n} \right\} \in \mathbb{R}^{(2n+1) \times N \times (3P^2)}$, where $N = HW/P^2$ is the number of tokens extracted from each frame with patches of size $P$. The visual encoder consists of token drop and feature extraction, as detailed below.

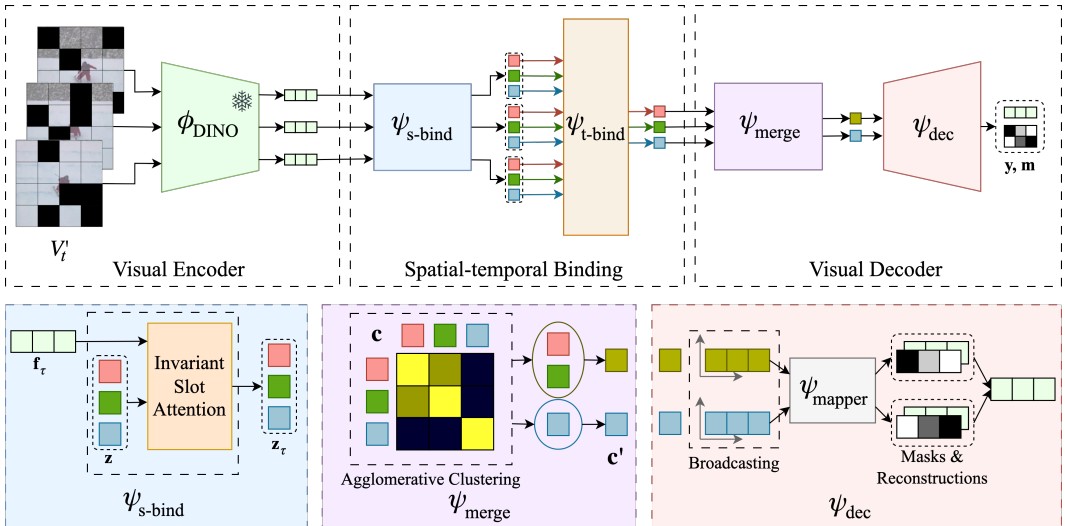

Figure 2: **Overview.** In this study, we introduce SOLV, an autoencoder-based model designed for object-centric learning in videos. Our model consists of three components: (i) **Visual Encoder** for extracting features for each frame using $\phi_{\text{DINO}}$; (ii) **Spatial-temporal Binding** module for generating temporally-aware object-centric representations by binding them spatially and temporally using $\psi_{\text{s-bind}}$ and $\psi_{\text{t-bind}}$, respectively; (iii) **Visual Decoder** for estimating segmentation masks and feature reconstructions for the central frame with $\psi_{\text{dec}}$, after merging similar slots using $\psi_{\text{merge}}$.

**Token Drop:** As input to the encoder, we only sample a subset of patches. Our sampling strategy is straightforward: we randomly drop the input patches with some ratio for each frame,

$$\mathcal{V}'_t = \left\{ \mathbf{v}'_{t-n}, \ldots, \mathbf{v}'_{t+n} \right\} = \left\{ \text{drop}\left(\mathbf{v}_{t-n}\right), \ldots, \text{drop}\left(\mathbf{v}_{t+n}\right) \right\} \in \mathbb{R}^{(2n+1) \times N' \times (3P^2)}, \quad N' < N \tag{2}$$

where $N'$ denotes the number of tokens after random sampling.

**Feature Extraction:** We use a frozen Vision Transformer (ViT) [14] with parameters initialized from DINOv2 [62], a self-supervised model that has been pretrained on a large number of images:

$$\mathcal{F} = \left\{ \mathbf{f}_{t-n}, \ldots, \mathbf{f}_{t+n} \right\} = \left\{ \phi_{\text{DINO}}\left(\mathbf{v}'_{t-n}\right), \ldots, \phi_{\text{DINO}}\left(\mathbf{v}'_{t+n}\right) \right\} \in \mathbb{R}^{(2n+1) \times N' \times D} \tag{3}$$

where $D$ refers to the dimension of output features from the last block of DINOv2, right before the final layer normalization.

**Discussion:** Our design of token drop serves two purposes: *Firstly*, masked autoencoding has been widely used in natural language processing and computer vision, acting as a proxy task for self-supervised learning, our token drop can effectively encourage the model to acquire high-quality visual representation; *Secondly*, for video processing, the extra temporal axis brings a few orders of magnitude of more data, processing sparsely sampled visual tokens can substantially reduce memory budget, enabling computation-efficient learning, as will be validated in our experiments.

### 3.2.2 Spatial-temporal Binding

After extracting visual features for each frame, we first spatially group the image regions into slots, with each specifying a semantic object, *i.e.*, discover objects within single image; then, we establish the temporal bindings between slots with a Transformer, *i.e.*, associate objects within video clips.

$$\Phi_{\text{st-bind}}\left(\mathcal{F}\right) = \psi_{\text{t-bind}}\left(\psi_{\text{s-bind}}\left(\mathbf{f}_{-n}\right), \ldots, \psi_{\text{s-bind}}\left(\mathbf{f}_{+n}\right)\right) \in \mathbb{R}^{K \times D_{\text{slot}}} \tag{4}$$

**Spatial Binding ($\psi_{\text{s-bind}}$):** The process of spatial binding is applied to each frame **independently**. We adopt the invariant slot attention proposed by Biza et al. [3], with one difference, that is, we use a shared initialization $\mathcal{Z}_\tau$ at each time step $\tau \in \{t-n, \ldots, t+n\}$. Specifically, given features after token drop at a time step $\tau$ as input, we learn a set of initialization vectors to translate and scale the input position encoding for each slot separately with $K$ slot vectors $\mathbf{z}^j \in \mathbb{R}^{D_{\text{slot}}}$, $K$ scale vectors

$\mathbf{S}_s^j \in \mathbb{R}^2$, $K$ position vectors $\mathbf{S}_p^j \in \mathbb{R}^2$, and one absolute positional embedding grid $\mathbf{G}_{\text{abs}} \in \mathbb{R}^{N \times 2}$. We mask out the patches corresponding to the tokens dropped in feature encoding (Section 3.2.1) and obtain absolute positional embedding for each frame $\tau$ as $\mathbf{G}_{\text{abs},\tau} = \text{drop}(\mathbf{G}_{\text{abs}}) \in \mathbb{R}^{N' \times 2}$. Please refer to the original paper [3] or the Supplementary Material for details of invariant slot attention on a single frame. This results in the following set of learnable vectors for each frame:

$$\mathcal{Z}_\tau = \left\{ \left( \mathbf{z}^j, \mathbf{S}_s^j, \mathbf{S}_p^j, \mathbf{G}_{\text{abs},\tau} \right) \right\}_{j=1}^K \tag{5}$$

Note that these learnable parameters are shared for all frames and updated with respect to the dense visual features of the corresponding frame. In other words, slots of different frames start from the same representation but differ after binding operation due to interactions with frame features. The output of $\psi_{\text{s-bind}}$ is the updated slots corresponding to independent object representations at frame $\tau$:

$$\{\mathbf{z}_\tau^j\}_{j=1}^K = \psi_{\text{s-bind}}(\mathbf{f}_\tau) \in \mathbb{R}^{K \times D_{\text{slot}}}, \quad \tau \in \{t-n, \ldots, t+n\} \tag{6}$$

In essence, given that consecutive frames typically share a similar visual context, the use of learned slots inherently promotes temporal binding, whereby slots with the same index can potentially bind to the same object region across frames. Our experiments demonstrate that incorporating invariant slot attention aids in learning slot representations that are based on the instance itself, rather than variable properties like size and location.

**Temporal Binding ($\psi_{\text{t-bind}}$):** Up until this point, the model is only capable of discovering objects by leveraging information from individual frames. This section focuses on how to incorporate temporal context to enhance the slot representation. Specifically, given the output slots from the spatial binding module, denoted as $\left\{ \{\mathbf{z}_{t-n}^j\}_{j=1}^K, \ldots, \{\mathbf{z}_{t+n}^j\}_{j=1}^K \right\} \in \mathbb{R}^{(2n+1) \times K \times D_{\text{slot}}}$, we apply a transformer encoder to the output slots with same index across different frames. In other words, the self-attention mechanism only computes a $(2n+1) \times (2n+1)$ affinity matrix across $2n+1$ time steps. This approach helps to concentrate on the specific region of the image through time. Intuitively, each slot learns about its future and past in the temporal window by attending to each other, which helps the model to create a more robust representation by considering representations of the same object at different times.

To distinguish between time stamps, we incorporate learnable temporal positional encoding onto the slots, *i.e.*, all slots from a single frame receive the same encoding. As the result of temporal transformer, we obtain the updated slots $\mathbf{c}$ at the target time step $t$:

$$\mathbf{c} = \Phi_{\text{st-bind}}(\mathcal{F}) \in \mathbb{R}^{K \times D_{\text{slot}}} \tag{7}$$

### 3.2.3 Visual Decoder

The spatial-temporal binding process yields a set of slot vectors $\mathbf{c} \in \mathbb{R}^{K \times D_{\text{slot}}}$ at target frame $t$ that are aware of temporal context. However, in natural videos, the number of objects within a frame can vary significantly, leading to over-clustering when a fixed number of slots is used, *i.e.*, as we can always initialize slots in an over-complete manner.

To overcome this challenge, we propose a simple solution for slot merging through Agglomerative Clustering. Additionally, we outline the procedure for slot decoding to reconstruct video features, which resembles a masked auto-encoder in feature space.

$$\Phi_{\text{vis-dec}}(\mathbf{c}) = \psi_{\text{dec}} \circ \psi_{\text{merge}}(\mathbf{c}) \tag{8}$$

**Slot Merging ($\psi_{\text{merge}}$):** As expected, the problem of object segmentation is often a poorly defined problem statement in the self-supervised scenario, as there can be multiple interpretations of visual regions. For instance, in an image of a person, it is reasonable to group all person pixels together or group the person's face, arms, body, and legs separately. However, it is empirically desirable for pixel embeddings from the same object to be closer than those of different objects. To address this challenge, we propose to merge slots using Agglomerative Clustering (AC), which does not require a predefined number of clusters. As shown in Fig. 3, we first compute the affinity matrix between all slots based on cosine similarity, then use this affinity matrix to cluster slots into groups, and compute the mean slot for each resulting cluster:

$$\mathbf{c}' = \psi_{\text{merge}}(\mathbf{c}) \in \mathbb{R}^{K_t \times D_{\text{slot}}}, \qquad K_t \leq K \tag{9}$$

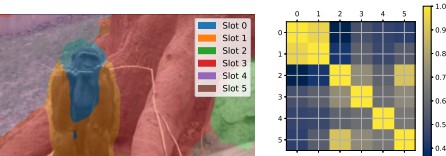

Figure 3: Assignments of slots to the input frame (**left**) and the pairwise similarity matrix of slots (**right**) where lighter colors indicate higher similarity. Slots corresponding to the parts of the same object (0 and 1) are highly similar to each other while being different from background slots (2, 3, 4, and 5).

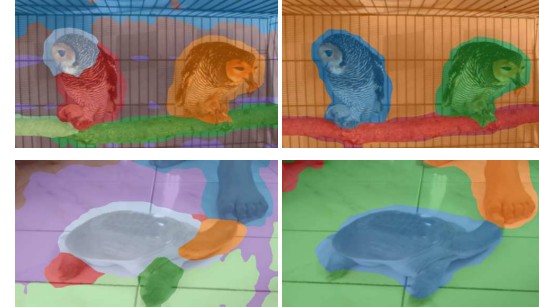

Figure 4: Results without (**left**) and with (**right**) slot merging.

By merging semantically similar slots that correspond to the same object, we can dynamically determine the optimal number of slots. Our slot merging strategy, as shown in Fig. 4, effectively combines regions of the same object using solely learned visual features. Note that slot merging is not a post-processing step, but rather an integral part of our training process.

**Decoder ($\psi_{\textbf{dec}}$):** We decode the merged slots $\mathbf{c}'$ with decoder $\psi_{\text{dec}}$ to obtain the corresponding segmentation mask $\mathbf{m}$ and the full reconstruction $\mathbf{y}$:

$$\mathbf{y}, \ \mathbf{m} = \psi_{\text{dec}} \left( \mathbf{c}' \right), \quad \mathbf{y} \in \mathbb{R}^{N \times D}, \ \mathbf{m} \in \mathbb{R}^{K_t \times N} \tag{10}$$

We reshape and upsample the masks $\mathbf{m}$ to the original input size to obtain the final segmentation. Similar to the MLP decoder design introduced in DINOSAUR [67], we use a spatial broadcast decoder [82] to reconstruct the **full** feature map, i.e., $\hat{\mathbf{y}}^j \in \mathbb{R}^{N \times D}$ of each slot $j$ with their alpha weights $\boldsymbol{\alpha}^j \in \mathbb{R}^N$. These alpha weights are converted to segmentation masks with a softmax. As per standard procedure, we add learned positional encodings to identify spatial locations during decoding. Each slot $\mathbf{c}'^j$, broadcasted to the shape of the input feature map, is decoded by a series of linear layers, $\psi_{\text{mapper}}$, with shared weights across all slots. The reconstruction is ultimately achieved through a weighted sum of the decoded slots:

$$\mathbf{y} = \sum_{j=1}^{K_t} \hat{\mathbf{y}}^j \odot \mathbf{m}^j, \qquad \mathbf{m}^j = \text{softmax} \left( \boldsymbol{\alpha}^j \right), \qquad \boldsymbol{\alpha}^j, \ \hat{\mathbf{y}}^j = \psi_{\text{mapper}} \left( \text{broadcast} \left( \mathbf{c}'^j \right) \right) \tag{11}$$

At training time, we optimise the model by minimising the difference between the feature map computed from original unmasked tokens of the frame at time $t$ and the reconstructed tokens $\mathbf{y}$:

$$\mathcal{L} = \| \phi_{\text{DINO}} \left( \mathbf{v}_t \right) - \mathbf{y} \|^2 \tag{12}$$

## 4 Experiments

### 4.1 Experimental Setup

**Datasets:** Our proposed method is evaluated on one synthetic and two real-world video datasets. For the synthetic dataset, we select MOVi [25], a widely-used benchmark for evaluating object-centric methods, particularly for multi-object segmentation in videos. To ensure a rigorous evaluation, we use the challenging MOVi-E dataset with up to 20 moving and static objects and random camera motion, as suggested by Bao et al. [2]. For the real-world datasets, we use the validation split of DAVIS17 [65]. In addition, we evaluate our method on a subset of the Youtube-VIS 2019 (YTVIS19) [87] train set, because there is no official validation or test set provided with ground-truth masks. Specifically, the first evaluation set consists of 30 videos out of 90 videos on DAVIS17, and the second evaluation set consists of 300 videos out of 2,883 high-resolution videos on YTVIS19. We perform all the ablations on the YTVIS19 dataset.

**Metrics:** For our synthetic dataset evaluation, we use the foreground adjusted rand index (FG-ARI) to measure the quality of clustering into multiple foreground objects. To maintain consistency with prior studies, we calculate the per-frame FG-ARI and report the mean across all frames, as done in

recent works such as Karazija et al. [36], Bao et al. [2], and Seitzer et al. [67]. This approach also allows us to compare to state-of-the-art image-based segmentation methods, such as DINOSAUR [67]. For real-world datasets, we use the mean Intersection-over-Union (mIoU) metric, which is widely accepted in segmentation, by considering only foreground objects. To ensure temporal consistency of assignments between frames, we apply Hungarian Matching between the predicted and ground-truth masks of foreground objects in the video, following the standard practice [65].

## 4.2 Results on Synthetic Data: MOVi-E

The results on the MOVi-E dataset are presented in Table 1, with methods divided based on whether they use an additional modality. For example, the sequential extensions of slot attention, such as SAVi [40] and SAVi++ [17], reconstruct different modalities like flow and sparse depth, respectively, their performance falls behind other approaches despite additional supervision. On the other hand, STEVE [69] improves results with a transformer-based decoder for reconstruction. PPMP [36] significantly improves performance by predicting probable motion patterns from an image with flow supervision during training. MoTok [2] outperforms other methods, but relies on motion segmentation masks to guide the attention maps of slots during training, with a significant drop in performance without motion segmentation. The impressive performance of DINOSAUR [67] highlights the importance of self-supervised training for segmentation. Our method stands out by using the spatial-temporal slot binding, and autoencoder training in the feature space, resulting in significantly better results than all previous work (see Fig. 5).

Table 1: **Quantitative Results on MOVi-E.** This table shows results in comparison to the previous work in terms of FG-ARI on MOVi-E.

| Model | +Modality | FG-ARI ↑ |
|---|---|---|
| SAVi [40] | Flow | 39.2 |
| SAVi++ [17] | Sparse Depth | 41.3 |
| PPMP [36] | Flow | 63.1 |
| MoTok [2] | Motion Seg. | 66.7 |
| MoTok [2] | | 52.4 |
| STEVE [70] | - | 54.1 |
| DINOSAUR [67] | | 65.1 |
| Ours | | **80.8** |



Figure 5: **Qualitative Results on MOVi-E.**

## 4.3 Results on Real Data: DAVIS17 and Youtube-VIS 2019

Due to a lack of multi-object segmentation methods evaluated on real-world data, we present a baseline approach that utilizes spectral clustering on the DINOv2 features [62]. To define the number of clusters, we conduct an oracle test by using the ground-truth number of objects. After obtaining masks independently for each frame, we ensure temporal consistency by applying Hungarian Matching between frames based on the similarity of the mean feature of each mask. However, as shown in Table 2, directly clustering the DINOv2 features produces poor results despite the availability of privileged information on the number of objects.

Table 2: **Quantitative Results on Real-World Data.** These results show the video multi-object evaluation results on the validation split of DAVIS17 and a subset of the YTVIS19 train split.

| Model | Supervision | DAVIS17 | | YTVIS19 | |
|---|---|---|---|---|---|
| | | mIoU ↑ | FG-ARI ↑ | mIoU ↑ | FG-ARI ↑ |
| DINOv2 [62] + Clustering | #Objects | 14.34 | 24.19 | 28.17 | 16.19 |
| OCLR [83] | Flow | **34.60** | 14.67 | 32.49 | 15.88 |
| Ours | - | 30.16 | **32.15** | **45.32** | **29.11** |

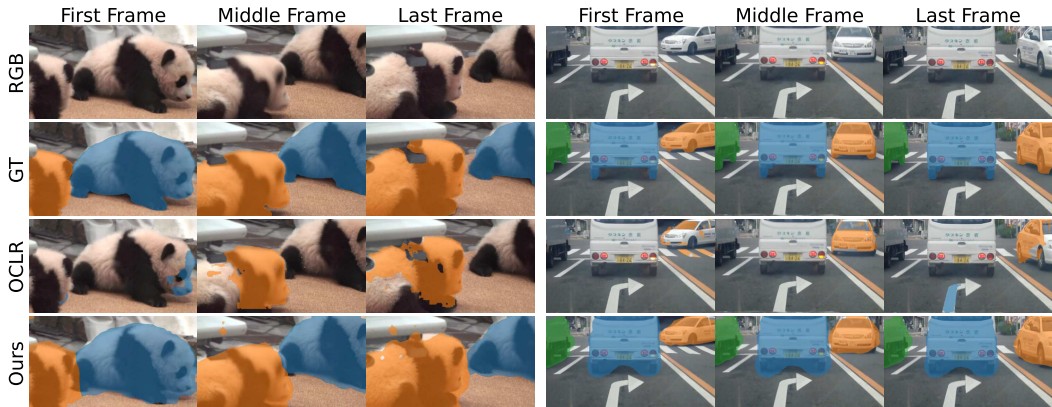

Figure 6: **Qualitative Results of Multi-Object Video Segmentation on YTVIS19.** We provide the first, middle, and last frames along with their corresponding segmentation after the Hungarian Matching step. Our model is proficient in accurately identifying objects in their entirety, irrespective of their deformability or lack of motion. This capability is highlighted in contrast to OCLR [83], whose results are displayed in the third row for comparison.

We also compare our method to the state-of-the-art approach OCLR [83], which is trained using a supervised objective on synthetic data with ground-truth optical flow. OCLR [83] slightly outperforms our method on DAVIS17, where the optical flow can often be estimated accurately, that greatly benefits the segmentation by allowing it to align with the real boundaries, resulting in better mIoU performance. However, our method excels at detecting and clustering multiple objects, even when the exact boundaries are only roughly located, as evidenced by the higher FG-ARI. The quality of optical flow deteriorates with increased complexity in YTVIS videos, causing OCLR [83] to fall significantly behind in all metrics. The qualitative comparison in Fig. 6 demonstrates that our method can successfully segment multiple objects in a variety of videos. Additional qualitative results can be found in the Supplementary Material.

### 4.4 Ablation Study

**On the Effectiveness of Architecture:** We conducted an experiment to examine the impact of each proposed component on the performance. *Firstly*, we replaced the spatial binding module, $\psi_{\text{s-bind}}$, with the original formulation in Slot Attention [52]. *Secondly*, we removed the temporal binding module, $\psi_{\text{t-bind}}$, which prevents information sharing between frames at different time steps. *Finally*, we eliminated the slot merging module, $\psi_{\text{merge}}$.

We can make the following observations from the results in Table 3: (i) as shown by the results of Model-A, a simple extension of DINOSAUR [67] to videos, by training with DINOv2 features and matching mask indices across frames, results in poor performance; (ii) only adding slot merging (Model-B) does not significantly improve performance, indicating that over-clustering is not the primary issue; (iii) significant mIoU gains can be achieved with temporal binding (Model-C) or

Table 3: **Components.** The effect of changing our spatial binding module ($\psi_{\text{s-bind}}$) to the original slot attention, removing the temporal binding module ($\psi_{\text{t-bind}}$) or the slot merging module ($\psi_{\text{merge}}$).

| Model | $\psi_{\text{merge}}$ | $\psi_{\text{s-bind}}$ | $\psi_{\text{t-bind}}$ | mIoU ↑ | FG-ARI ↑ |
|---|---|---|---|---|---|
| A | ✗ | ✗ | ✗ | 37.75 | 27.05 |
| B | ✓ | ✗ | ✗ | 38.23 | 29.09 |
| C | ✓ | ✗ | ✓ | 44.95 | 28.42 |
| D | ✓ | ✓ | ✗ | 44.94 | 27.38 |
| E | ✓ | ✓ | ✓ | **45.32** | **29.11** |

Table 4: **Number of Slots.** The effect of varying the number of slots with/without slot merging.

| #Slots | Merging | mIoU ↑ | FG-ARI ↑ |
|---|---|---|---|
| 6 | ✗ | 43.29 | 27.73 |
|  | ✓ | 44.90 | 28.52 |
| 8 | ✗ | 39.90 | 22.55 |
|  | ✓ | **45.32** | **29.11** |
| 12 | ✗ | 36.04 | 20.20 |
|  | ✓ | 43.19 | 27.94 |

Table 5: **Visual Encoder**. The effect of different types of self-supervised pretraining methods and architectures for feature extraction.

| Model | mIoU ↑ | FG-ARI ↑ |
|---|---|---|
| Supervised ViT-B/16 | 37.58 | 19.94 |
| DINO ViT-B/8 | 39.53 | 24.70 |
| DINO ViT-B/16 | 41.91 | 24.01 |
| DINOv2 ViT-B/14 | **45.32** | **29.11** |

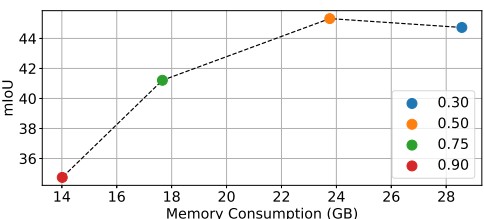

Figure 7: **Token Drop Ratio.** The effect of varying token drop ratio on performance (mIoU) and memory consumption.

spatial binding (Model-D), which shows the importance of spatially grouping pixels within each frame as well as temporally relating slots in the video. Finally, the best performance is obtained with Model-E in the last row by combining all components.

**Number of Slots:** We perform an experiment by varying the number of slots with and without slot merging in Table 4. Consistent with the findings in previous work [67], the performance is highly dependent on the number of slots, especially when slot merging is not applied. For example, without slot merging, increasing the number of slots leads to inferior results in terms of both metrics, due to the over-segmentation issue in some videos. While using slot merging, we can better utilize a larger number of slots as can be seen from the significantly improved results of 8 or 12 slots with slot merging. This is also reflected in FG-ARI, pointing to a better performance in terms of clustering. We set the number of slots to 8 and use slot merging in our experiments. We provide a visual comparison of the resulting segmentation of an image with and without slot merging in Fig. 4. Our method can successfully merge parts of the same object although each part is initially assigned to a different slot.

**Token Drop Ratio:** We ablated the ratio of tokens to drop in Fig. 7. As our motivation for the token drop is two-fold (Section 3.2.1), we show the effect of varying the token drop ratio on both performance and memory usage. Specifically, we plot mIoU on the vertical axis versus the memory consumption on the horizontal axis. We report the memory footprint as the peak in the GPU memory usage throughout a single pass with constant batch size. Fig. 7 confirms the trade-off between memory usage and performance. As the token drop ratio decreases, mIoU increases due to less information loss from tokens dropped. However, this comes at the cost of significantly more memory usage. We cannot report results by using all tokens, *i.e.* token drop ratio of 0, due to memory constraints. Interestingly, beyond a certain threshold, *i.e.* at 0.5, the token drop starts to act as regularization, resulting in worse performance despite increasing the number of tokens kept. We use 0.5 in our experiments, which reaches the best mIoU with a reasonable memory requirement. In summary, the token drop provides not only efficiency but also better performance due to the regularization effect.

**Visual Encoder:** We conducted experiments to investigate the effect of different visual encoders in Table 5. We adjust the resize values to maintain a consistent token count of 864 for different architectures. These findings underscore the critical role of pretraining methods. Please see Supplementary for a qualitative comparison. Notably, the DINOv2 model [62] outperforms its predecessor, DINO [7], in terms of clustering efficiency. On the other hand, models pretrained using supervised methods results in the weakest performance, with a substantial gap in the FG-ARI metric when compared to self-supervised pretraining methods. Furthermore, no specific patch size offers a clear advantage within the same pretraining type. For instance, DINO pretraining with ViT-B/8 exhibits superior performance in the FG-ARI metric but lags behind its 16-patch-sized counterpart in the mIoU metric.

## 5 Discussion

We presented the first fully unsupervised object-centric learning method for scaling multi-object segmentation to in-the-wild videos. Our method advances the state-of-the-art significantly on commonly used simulated data but more importantly, it is the first fully unsupervised method to report state-of-the-art performance on unconstrained videos of the Youtube-VIS dataset.

This work takes a first step towards the goal of decomposing the world into semantic entities from in-the-wild videos without any supervision. The performance can always be improved with better

features from self-supervised pretraining as shown in the comparison of different DINO versions. While we obtain significant performance boosts from self-supervised pretraining, it also comes with some limitations. For example, our method can locate objects roughly, but it fails to obtain pixel-accurate boundaries due to features extracted at patch level. Furthermore, nearby objects might fall into the same patch and cause them to be assigned to the same slot. Our slot merging strategy, although has been proved effective as a simple fix for the issue of over-segmentation, remains not differentiable. Future work needs to go beyond that assumption and develop methods that can adapt to a varying number of objects in a differentiable manner.

## 6  Acknowledgements

Weidi Xie would like to acknowledge the National Key R&D Program of China (No. 2022ZD0161400).

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

# Appendices

In this supplementary material, we discuss the broader impact (Section A), report experiment details including dataset, model, and training details (Section B), provide more details of Invariant Slot Attention, as mentioned in the main paper (Section C), perform extra ablation studies on input resolution and generalization performance of our model (Section D), compare the performance to supervised models (Section E), provide an analysis of failure cases (Section F), and finally present extra visualizations (Section G). In the last section, we provide a visual comparison between models in our ablation studies as well as previous work.

## A  Broader Impact

We proposed an object-oriented approach that can be applied to videos of real-world environments. We trained our model on the Youtube-VIS 2019 dataset that includes videos of humans. The proposed work can be used to locate and segment multiple instances of a wide variety of objects including all kinds of animals and humans from videos. While progress in this area can be used to improve not only human life but also, for example, wildlife, we acknowledge that it could also inadvertently assist in the creation of computer vision applications that could potentially harm society.

## B  Experiment Details

### B.1  Dataset Details

We eliminated the black borders for all videos in YTVIS19 dataset. Since we propose a self-supervised model, we merge the available splits on datasets for training. For all datasets except YTVIS, we use the available validation splits for evaluation. The annotations for the validation split are missing on the YTVIS, therefore, we use a subset of 300 videos from the train split for evaluation. We will share the indices of selected videos for future comparisons together with the code. For evaluation, we upsample the segmentation masks to match the resolution of the original input frames by using bilinear interpolation.

### B.2  Model Details

**Feature Extractor ($\phi_{\textbf{DINO}}$):**  We use the ViT-B/14 architecture with DINOv2 pretraining [62] as our default feature extractor. Our feature vector is the output of the last block without the CLS token. We add positional embeddings to the patches and then drop the tokens.

**Spatial Binding($\psi_{\textbf{s-bind}}$):**  We project the feature tokens from $\phi_{\text{DINO}}$ to slot dimension $D_{\text{slot}} = 128$, with a 2-layer-MLP, followed by layer normalization. Then the slots and the projected tokens are passed to the Invariant Slot Attention (ISA) (as detailed in Section C) as input. After slot attention, slots are updated with a GRU cell. Following the sequential update, slots are passed to a residual MLP with a hidden size of $4 \times D_{\text{slot}}$. All projection layers ($p, q, k, v, g$) have the same size as slots, *i.e.* $D_{\text{slot}}$. We repeat the binding operation 3 times. We multiply the scale parameter $\mathbf{S}_s$ by $\delta = 5$ to prevent relative grid $\mathbf{G}_{\text{rel}}$ from containing large numbers.

We use the following initializations for the learnable parameters: $\mathbf{G}_{\text{abs}}$, a coordinate grid in the range $[-1, 1]$; slots $\mathbf{z}$, Xavier initialization [24]; slot scale $\mathbf{S}_s$ and slot position $\mathbf{S}_p$, a normal distribution.

**Temporal Binding ($\psi_{\textbf{t-bind}}$):**  We use a transformer encoder with 3 layers and 8 heads [74] for temporal binding. The hidden dimension of encoder layers is set to $4 \times D_{\text{slot}}$. We initialize the temporal positional embedding with a normal distribution. We masked the slots of not available frames, *i.e.* frames with indices that are either less than 0 or exceed the frame number, in transformer layers.

**Slot Merging ($\psi_{\textbf{merge}}$):**  We use the implementation of Agglomerative Clustering in the `sklearn` library [64] with complete linkage. For each cluster, we compute the mean slot and the sum of attention matrices, *i.e.* matrix $\mathbf{A}$ in (13), for the associated slots. We determine the scale $\mathbf{S}_s$ and position $\mathbf{S}_p$ parameters for the merged attention values to calculate the relative grid $\mathbf{G}_{\text{rel}}$ of the new

slots. Then, $\mathbf{G}_{\text{rel}}$ is projected onto $D_{\text{slot}}$ using a linear layer $h$ and added to the broadcasted slots before decoding.

**Decoder Mapper ($\psi_{\textbf{mapper}}$):** The mapper $\psi_{\text{mapper}}$ consists of 5 linear layers with ReLU activations and a hidden size of 1024. The final layer maps the activations to the dimension of ViT-B tokens with an extra alpha value *i.e.* 768 + 1.

### B.3  Training Details

In all our experiments, unless otherwise specified, we employ DINOv2 [62] with the ViT-B/14 architecture. We set the number of consecutive frame range $n$ to 2 and drop half of the tokens before the slot attention step. We train our models on $2\times$V100 GPUs using the Adam [39] optimizer with a batch size of 48. We clip the gradient norms at 1 to stabilize the training. We match mask indices of consecutive frames by applying Hungarian Matching on slot similarity to provide temporal consistency. To prevent immature slots in slot merging, we apply merging with a probability that is logarithmically increasing through epochs.

**MOVi-E:** We train our model from scratch for a total of 60 epochs, which is equivalent to approximately 300K iterations. We use a maximum learning rate of $4 \times 10^{-4}$ and an exponential decay schedule with linear warmup steps constituting 5% of the overall training period. The model is trained using 18 slots and the input frames are adjusted to a size of $336 \times 336$, leading to 576 feature tokens for each frame. The slot merge coefficient in $\psi_{\text{merge}}$ is configured to 0.12.

**YTVIS19:** Similar to MOVi-E, we train the model from scratch for 180 epochs, corresponding to approximately 300K iterations with a peak learning rate of $4 \times 10^{-4}$ and decay it with an exponential schedule. Linear warmup steps are introduced for 5% of the training timeline. The model training involves 8 slots, and the input frames are resized to dimensions of $336 \times 504$, resulting in 864 feature tokens for each frame. The slot merge coefficient in $\psi_{\text{merge}}$ is set to be 0.12.

**DAVIS17:** Due to the small size of DAVIS17, we fine-tune the model pretrained on the YTVIS19 dataset explained above. We finetune on DAVIS17 for 300 epochs, corresponding to approximately 40K iterations with a reduced learning rate of $1 \times 10^{-4}$. We use the same learning rate scheduling strategy as in YTVIS19. During the fine-tuning process, we achieve the best result without slot merging, likely due to the fewer number of objects, typically one object at the center, on DAVIS17 compared to YTVIS19.

## C  Invariant Slot Attention

In this section, we provide the details of invariant slot attention (ISA), initially proposed by Biza et al. [3]. We use ISA in our Spatial Binding module $\psi_{\text{s-bind}}$ with shared initialization as explained in the main paper. Given the shared initialization $\mathcal{Z}_\tau = \left\{ \left( \mathbf{z}^j, \mathbf{S}_s^j, \mathbf{S}_p^j, \mathbf{G}_{\text{abs},\tau} \right) \right\}_{j=1}^K$, our goal is to update the $K$ slots: $\{ \mathbf{z}^j \}_{j=1}^K$. For clarification, in the following, we focus on the computation of single-step slot attention for time step $\tau$:

$$\mathbf{A}^j := \operatorname*{softmax}_K \left( \mathbf{M}^j \right) \in \mathbb{R}^{N'}, \qquad \mathbf{M}^j := \frac{1}{\sqrt{D_{\text{slot}}}} p \left( k \left( \mathbf{f}_\tau \right) + g \left( \mathbf{G}_{\text{rel},\tau}^j \right) \right) q \left( \mathbf{z}^j \right)^T \in \mathbb{R}^{N'} \quad (13)$$

where $p$, $g$, $k$, and $q$ are linear projections while the relative grid of each slot is defined as:

$$\mathbf{G}_{\text{rel},\tau}^j := \frac{\mathbf{G}_{\text{abs},\tau} - \mathbf{S}_p^j}{\mathbf{S}_s^j} \in \mathbb{R}^{N' \times 2} \tag{14}$$

The slot attention matrix $\mathbf{A}$ from (13) is used to compute the scale $\mathbf{S}_s$ and the positions $\mathbf{S}_p$ of slots following Biza et al. [3]:

$$\mathbf{S}_s^j := \sqrt{\frac{\operatorname{sum}\left( \mathbf{A} \odot \left( \mathbf{G}_{\text{abs},\tau} - \mathbf{S}_p^j \right)^2 \right)}{\operatorname{sum}\left( \mathbf{A}^j \right)}} \in \mathbb{R}^2, \qquad \mathbf{S}_p^j := \frac{\operatorname{sum}\left( \mathbf{A}^j \odot \mathbf{G}_{\text{abs},\tau} \right)}{\operatorname{sum}\left( \mathbf{A}^j \right)} \in \mathbb{R}^2 \quad (15)$$

After this step, following the original slot attention, input features are aggregated to slots using the weighted mean with another linear projection $v$:

$$\mathbf{U} := \mathbf{W}^T p \left( v \left( \mathbf{f}_\tau \right) + g \left( \mathbf{G}_{\text{rel},\tau}^j \right) \right) \in \mathbb{R}^{K \times D_{\text{slot}}}, \qquad \mathbf{W}^j := \frac{\mathbf{A}^j}{\operatorname{sum}\left( \mathbf{A}^j \right)} \in \mathbb{R}^{N'} \quad (16)$$

Then, $\mathbf{U}$ from (16) is used to update slots $\{\mathbf{z}^j\}_{j=1}^K$ with GRU followed by an additional MLP as residual connection as shown in (17). This operation is repeated 3 times.

$$\mathbf{z} := \mathbf{z} + \text{MLP}\left((\text{norm}(\mathbf{z}))\right), \qquad \mathbf{z} := \text{GRU}\left(\mathbf{z}, \mathbf{U}\right) \in \mathbb{R}^{K \times D_{\text{slot}}} \tag{17}$$

## D Additional Ablation Studies

In this section, we provide additional experiments to show the effect of resolution on the segmentation quality. We also report the results by varying the evaluation split on the YTVIS to confirm the generalization capability of our model.

**Effect of Resolution:** We conducted experiments to investigate the effect of the input frame resolution, *i.e.* the number of input tokens, in Fig. 8. Specifically, we experimented with resolutions of $168 \times 252$, $224 \times 336$, and our default resolution $336 \times 506$, corresponding to 216, 384, and 864 input feature tokens, respectively. These experiments show that the input resolution is crucial for segmentation performance.

**Varying The Evaluation Split:** As stated before, we cannot use the validation split of YTVIS19 due to manual annotations that are not publicly available. For evaluation, we only choose a subset of 300 videos. Here, we perform an experiment to examine the effect of varying the set of evaluation videos on performance.

We repeat the experiment in Table 3 by choosing mutually exclusive subsets of 300 videos, 3 times. We report mean and standard deviation ($\mu \pm \sigma$) of experiments for each model in Table 6. These results are coherent with the reported result on the fixed subset, showing that segmentation performance peaks when all of our components are combined, *i.e.* model E. Similarly, removing our components $\psi_{\text{t-bind}}$ (model D) and $\psi_{\text{s-bind}}$ (model C) one at a time leads to a performance drop, as shown in Table 3. Removing both (model B), results in the worst performance, even falling behind its counterpart without slot merging (model A). Overall, the results of varying the evaluation set agree with the reported performance on the selected subset in the main paper. This shows that the performance of our model generalizes over different evaluation subsets.

Table 6: **Varying evaluation splits.** The effect of changing evaluation sets for models in the component ablation.

| Model | mIoU ↑ | FG-ARI ↑ |
|-------|--------|----------|
| A | $37.72 \pm 0.82$ | $29.06 \pm 2.24$ |
| B | $37.20 \pm 1.53$ | $30.15 \pm 1.90$ |
| C | $42.76 \pm 2.09$ | $30.03 \pm 1.67$ |
| D | $42.98 \pm 2.01$ | $29.69 \pm 2.02$ |
| E | $43.28 \pm 2.57$ | $31.33 \pm 1.51$ |

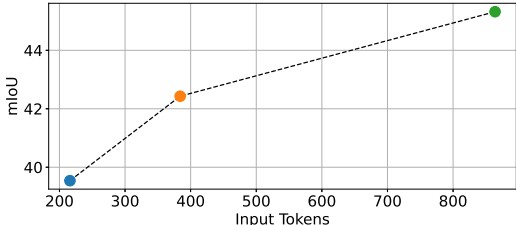

Figure 8: **Input resolution.** The effect of input resolution, *i.e.* the number of tokens, on performance (mIoU).

## E Additional Comparisons

Table 7: **Comparison to supervised models.** These results show the video multi-object evaluation results on the validation split of DAVIS17 between our model and supervised models.

| Model | Unsupervised | mIoU ↑ |
|-------|--------------|--------|
| UnOVOST [55] | ✗ | 66.4 |
| Propose-Reduce [48] | ✗ | 67.0 |
| Ours | ✓ | 30.2 |

We compare the performance of our model to available supervised models, exploiting pre-trained object detectors on large data in a supervised manner, for unsupervised video object segmentation on DAVIS17 in Table 7. The results underscore a notable gap between supervised and unsupervised approaches.

# F  Failure Analysis

Although our model can detect in-the-wild objects in different scales, segmentation boundaries are not perfectly aligned with the object due to patch-wise segmentation, as has been pointed out in the Discussion (Section 5). In addition to these observations, we identify three types of commonly occurring failure cases: (i) The over-clustering issue that remains unresolved in some cases even with slot merging. (ii) The tendency to cluster nearby instances of the same class into a single slot. (iii) Failure to detect small objects, particularly when they are situated near large objects. We provide visual examples of these cases in Fig. 9.

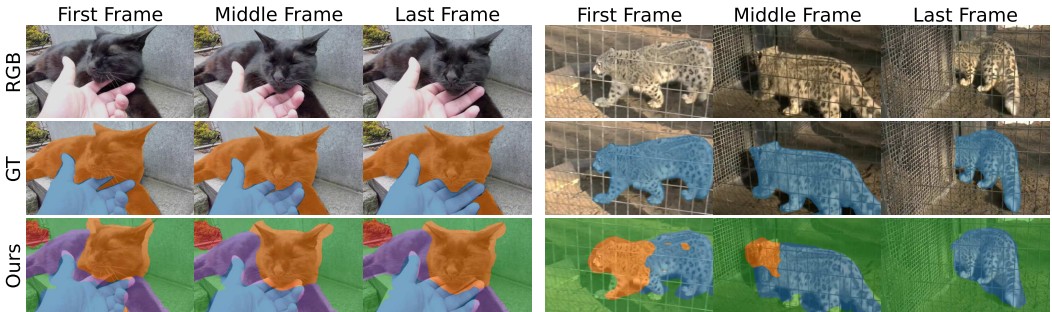

(a) Failure cases due to over-clustering.

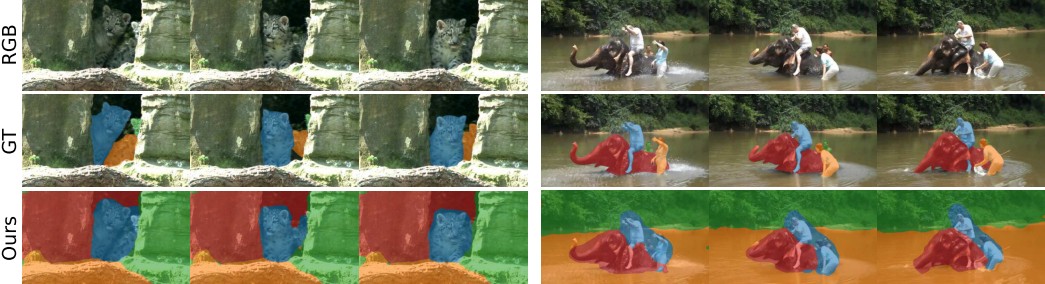

(b) Failure cases due to grouping nearby instances of the same class into one cluster.

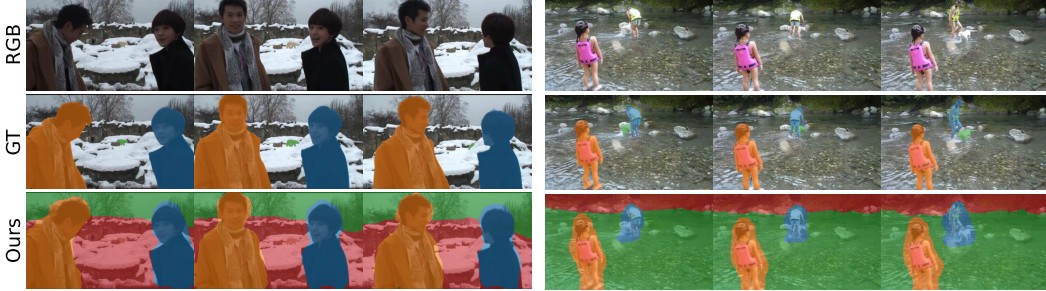

(c) Failure cases due to missing relatively small objects.

Figure 9: Failure cases of our model with potential reasons, grouped into three.

# G  Additional Visualizations

We provide additional qualitative results in Fig. 10. Our model can recognize not only the most salient object in the middle but also small, subtle objects in the background. Furthermore, it can handle a varying number of objects in the scene with slot merging.

**Feature Extractor:** In Fig. 11, we provide visualizations of different feature extractors, corresponding to the quantitative evaluation in Table 5, including DINOv2 [62], DINO [7], and Supervised. Self-supervised models performs better than the supervised one, also qualitatively. In particular, DINOv2 stands out for its exceptional capability to learn object representations across a diverse range

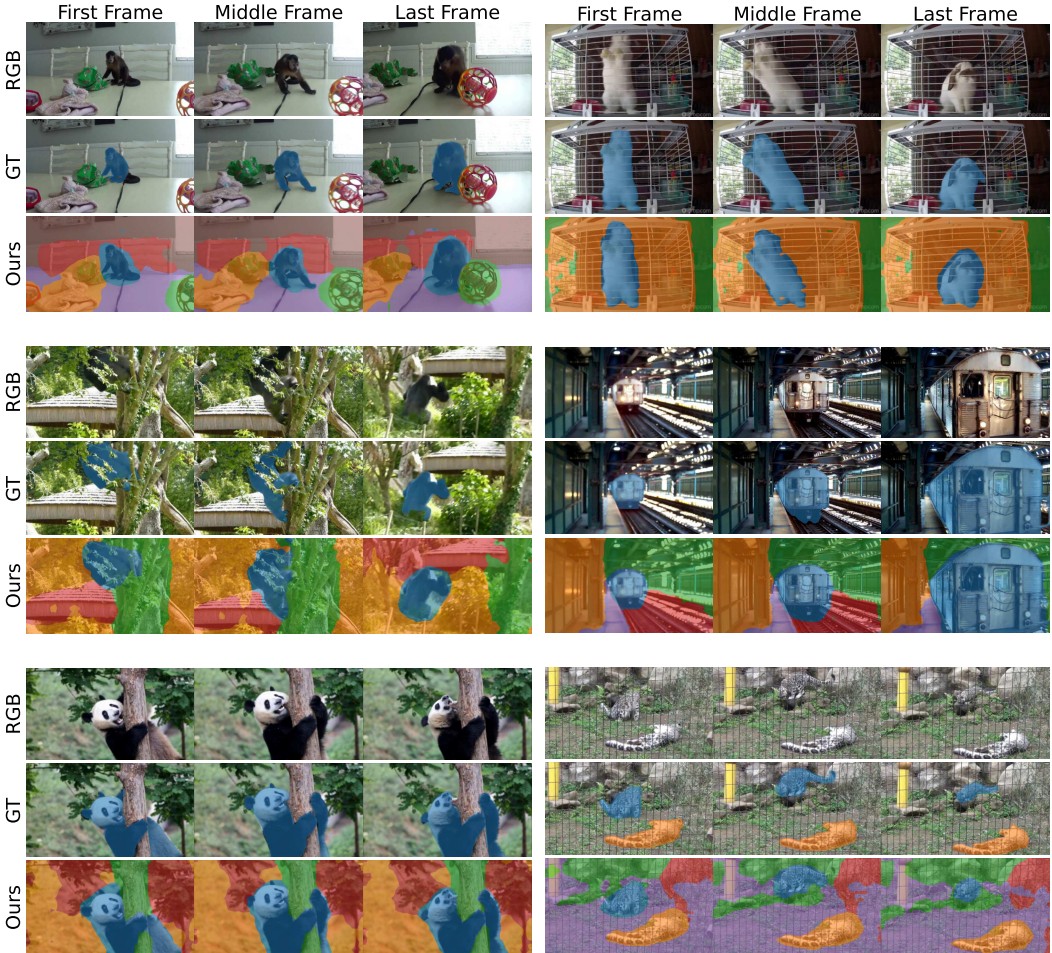

Figure 10: Qualitative results of multi-object video segmentation on YTVIS19

of categories. It also effectively identifies and segments intricate details that are missed by other feature extractors such as tree branches on the left and the bag on the table on the right.

**Components:** In Fig. 12, we visually compare the segmentation results of the models corresponding to the component ablation study in the main paper (Table 3 ). First of all, model A, which corresponds to the temporally consistent DINOSAUR [67], struggles to cluster the instances as a whole and also fails to track all objects, for instance, the human on the right, due to discrepancies in mask index across three frames. With the help of slot merging, model B effectively addresses the over-clustering issue, as observed in the mask of the calf on the left. Integrating our binding modules $\psi_{\text{t-bind}}$ and $\psi_{\text{s-bind}}$, resulting in model C and D, respectively, leads to a marked improvement in both segmentation and tracking quality. On the other hand, both models C and D have shortcomings in detecting the human in certain frames of the right video. Finally, combining them, our full model, *i.e.* model E, excels at segmenting and tracking not only labeled objects but also other objects, such as the car visible in the first frame of the second video.

**Comparison:** We provide a visual comparison between OCLR [83] and our model in Fig. 13 after matching the ground-truth and predictions. OCLR fails to detect static objects (top-left, middle-left, bottom-left) and deformable objects (middle-right) due to failures of optical flow in these cases. Moreover, it considers moving regions as different objects such as the water waves (top-right). On the other hand, SOLV can accurately detect all objects as a whole.

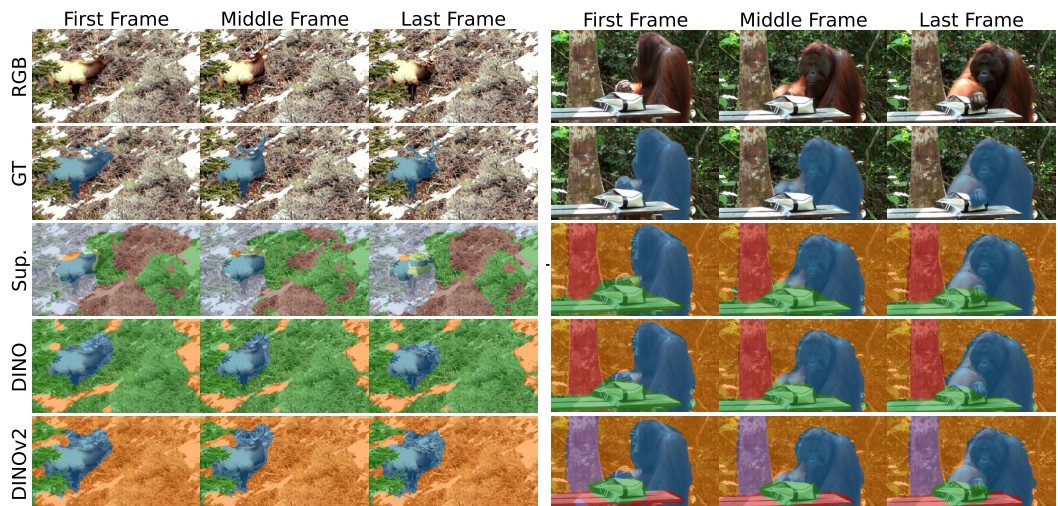

Figure 11: Qualitative comparison of different pretraining methods for visual encoder.

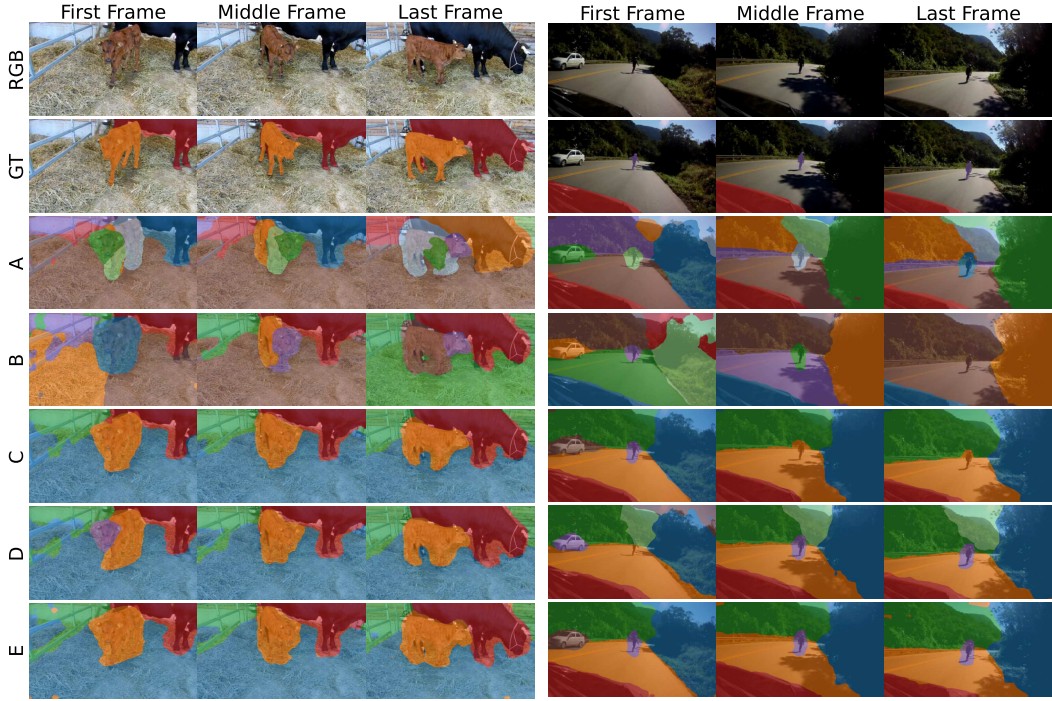

Figure 12: Qualitative results of different models in the component ablation study (Table 3).

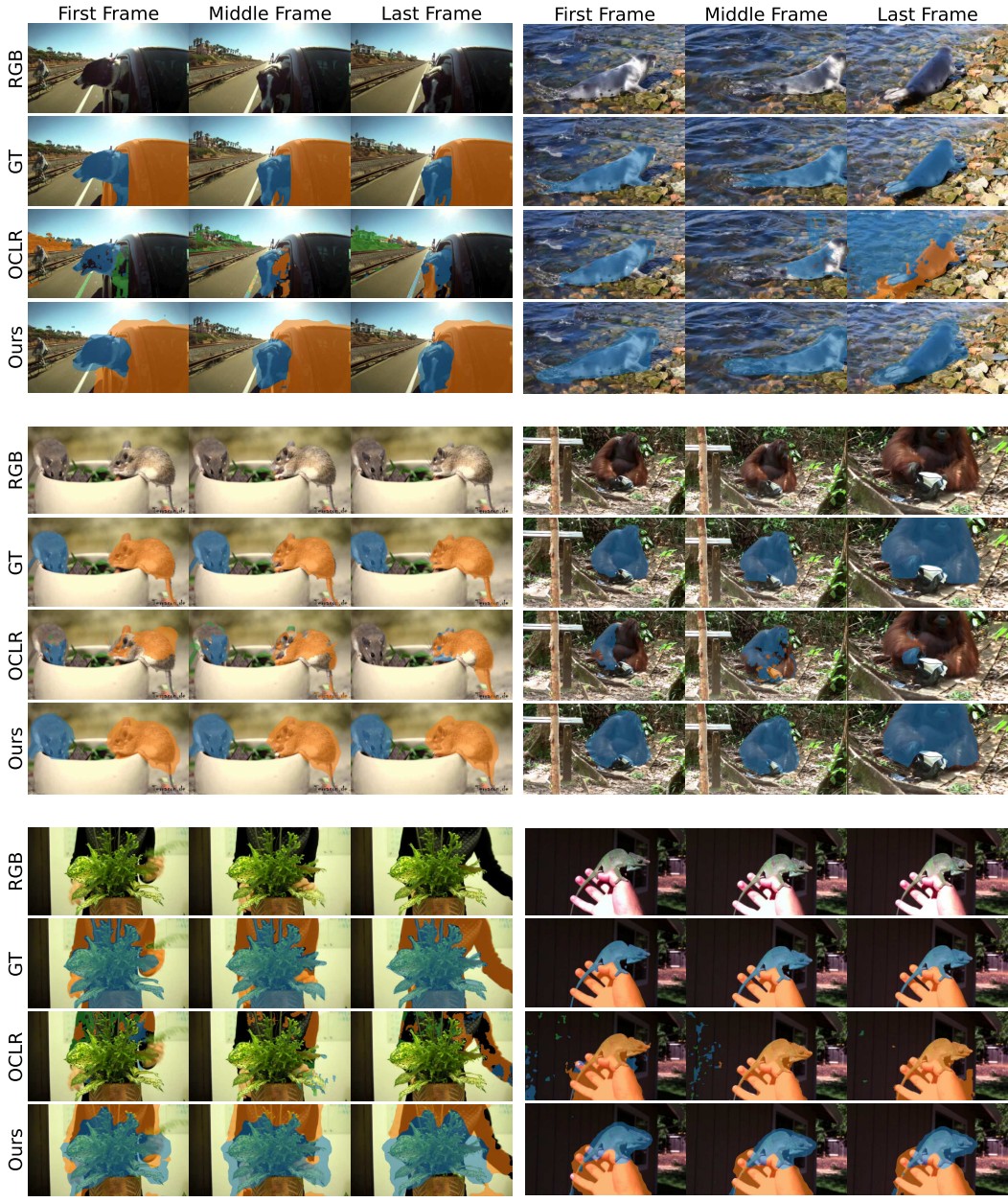

Figure 13: Qualitative results of multi-object video segmentation on YTVIS19 after Hungarian Matching is applied. Segmentation results of OCLR [83] are provided in third row.

