# OpenReview forum: "Self-supervised Object-Centric Learning for Videos"
_NeurIPS.cc/2023/Conference — NeurIPS 2023 poster_

### Official Review · Reviewer_zEVX · 2023-07-05

**Soundness:** 3 good
**Presentation:** 3 good
**Contribution:** 3 good
**Rating:** 7
**Confidence:** 3

**Summary:**

This paper introduced a fully unsupervised method, SOLV, for segmenting multiple objects in real-world sequences. In this paper, the author employed the token drop strategy to reduce computation and enhance regularization. Additionally, Spatial-temporal Binding was proposed to aggregate the features of objects within and across frames, resulting in a remarkable improvement in mIoU. To address over-clustering, the author employed Agglomerative Clustering to merge slots. It is noteworthy that SOLV not only significantly advances the state-of-the-art on commonly used simulated data but also stands out as the first fully unsupervised method to demonstrate state-of-the-art performance on unconstrained videos from the Youtube-VIS dataset.

**Strengths:**

1. A fully unsupervised object-centric learning method for scaling multi-object segmentation to in-the-wild videos was first proposed and achieved significant improvements on traditional synthetic datasets. Moreover, it has delivered impressive results on real-world video datasets.
2. Many details in the method, such as token drop ratio, number of slots, and variations of the visual encoder, have undergone ablation experiments, enhancing the robustness and reproducibility of the work.
3. The slot merging strategy significantly reduces the sensitivity of the model's performance to the number of slots.
4. Both temporal binding and spatial binding techniques bring significant gains in mIoU.

**Weaknesses:**

1. A more elaborate explanation is required to clarify how the Agglomerative Clustering algorithm is employed to dynamically determine the optimal number of slots.

**Questions:**

1. In the ablation experiments, a notable observation arises regarding the performance of Model-C and Model-D compared to their predecessor, Model-B, as they exhibit a decrease in FG-ARI. However, Model-E, which combines the spatial bind and temporal bind components from both Model-C and Model-D, ultimately achieved superior results in terms of FG-ARI. What are the possible reasons behind this phenomenon？

2. Whether the shape of $\mathcal V_t$ at line 120 is indeed $\mathbb R^{(2n+1)\times N \times (3P^2)}$, and the dimension of $\mathcal V_t'$ in Equation (2) should be $\mathbb R^{(2n+1)\times N \times (3P^2)}$.


**Limitations:**

1. As discussed at the end of the paper, it is acknowledged that Agglomerative Clustering is non-differentiable during the training process. Therefore, it might be worthwhile to explore alternative clustering algorithms as potential alternatives.
2. Based on the visualized results, it is evident that SOLV excels in the detection and clustering of foreground objects. However, there is room for improvement in terms of the accuracy of segmentation boundaries.

---

> ### Author Rebuttal · Authors · 2023-08-09
>
> We extend our sincere appreciation to the reviewer for their valuable insights and constructive feedback. We have responded to their questions and concerns, and we hope our explanations thoughtfully encompass the raised points.
>
> ## W1. Explanations on slot merging
> Please refer to Q4 of the global response.
>
> ## Q1. Reasons behind the model behaviours in the ablation
> Please refer to Q2 of the global response.
>
> ## Q2. Typo in the text
> We appreciate the reviewer for their extremely careful reading and inspection. We will add the missing patch size in line 120.

---

> > ### Comment · Reviewer_zEVX · 2023-08-20
> >
> > Thank authors for the rebuttal. It solves my concerns. I keep my original rating as accept.

---

### Official Review · Reviewer_gzBP · 2023-07-06

**Soundness:** 3 good
**Presentation:** 3 good
**Contribution:** 2 fair
**Rating:** 5
**Confidence:** 4

**Summary:**

The paper introduces an approach to segment multiple objects for video sequence in both real and synthetic data without utilizing any additional signals besides RGB frames. It has 3 components including a visual encoder, a spatial-temporal binding model used for grouping pixels into slots across different time, and a visual decoder to get the segmentation mask. The method also uses a slot merging strategy to address the over-segmentation issue caused by a fixed number of slots. The paper conducts experiments and it outperforms previous methods on synthetic and real data. The ablation studies are also presented to prove each component's effectiveness.

**Strengths:**

This paper can segment complex scenes in Youtube videos and achieve satisfying qualitative results.
It didn’t rely on extra signals in videos which is claimed not stable in video segmentation.
The drop approach can save memory and calculation time successfully.
This paper achieves great performance on MOVi-E.

**Weaknesses:**

This paper didn't test on single object segmentation dataset like DAVIS 2017 and compare methods like CIS, Y Yang 2019.

**Questions:**

For the video segmentation dataset like DAVIS17, how could you choose which segment mask as the object mask, not the background, used to calculate the metric?

How to make sure that the merging mechanism can successfully merge the masks into a whole object? How to determine which parts should be merged into one object?

**Limitations:**

It could be better to visualize several cases of each slot's segmentation mask results for slot=8 which is claimed to be the best in Table 4.

---

> ### Author Rebuttal · Authors · 2023-08-09
>
> We appreciate the feedback and valuable comments you have shared. In response, we have furnished explanations, aiming to address your concerns effectively, hoping the scores can be raised accordingly.
>
> ## W1. Single object segmentation
> We did test DAVIS2017 in our paper, the results are reported in Table 2, it is a dataset of multi-object video segmentation. In terms of single video object segmentation, existing datasets are usually defined for motion segmentation, that assigns all pixels with similar motion into one group, regardless of semantic categories.
>
> While in our case, we aim to segment objects based on semantics, for example, in the given example from DAVIS17 (Fig 14 (a) in FD), the person and the bicycle are segmented into two different groups, consistent with the ground-truth (Fig 14 (b) in FD) and mIoU is calculated as 60.0. However, when ground-truth annotations are turned into single-object evaluation (Fig 14 (c) in FD), mIoU score drops to 41.2 (for the whole video).
>
> Nevertheless, we provide the results of our models on DAVIS17, which is designed for multi object segmentation:
> - 6 slots, no merging: 39.06 mIoU
> - 8 slots, merging: 47.50 mIoU
> - 4 slots, merging: 49.01 mIoU
> - CIS [1]: 53.1 mIoU
> - DyStaB [2]: 58.9 mIoU
>
> ## Q1. Mask matching for evaluation
> We follow the common practice in this community, and use Hungarian Matching for evaluation, as also used in DINOSAUR, SAVi, OCLR. We will clarify this evaluation metric in our final paper.
>
> ## Q2. Slot merging mechanism
> Please refer to Q4 of the global response.
>
> ## L1. Qualitative examples
> We have provided additional qualitative results of our best model for masks of each slot in Figure 9 of Supplementary. Note that, although the model is trained with 8 slots, some cases result in fewer masks due to dynamically set number of slots in slot merging.
>
>
> **References**
>
> [1] Yang et al., Unsupervised Moving Object Detection via Contextuaş Information Separation, CVPR19
> [2] Yang et al., DyStaB: Unsupervised Object Segmentation via Dynamic-Static Bootstrapping, CVPR21

---

> > ### Comment · Reviewer_gzBP · 2023-08-18
> >
> > Thank you very much for your clarification which addresses my lots of concerns. While I still have some questions:
> > (1) Why when ground-truth annotations are turned into single-object evaluation the mIoU score drops to 41.2 from 60.0? Could you please provide any insights or analysis about this?
> > (2) As reiviewer xws8 referred that this method seem not to be able to segment the accurate object boundary, I understand this is due to the algorithm runs at patch-level image token input but how could this weakness be improved?

---

> > > ### Author Response · Authors · 2023-08-19
> > >
> > > Thank you for your insights and the questions raised.
> > >
> > > (i) The drop in mIoU when transitioning from multi-object to single-object evaluation is due to the nature of the evaluation protocol:
> > > * The single-object evaluation is on **motion segmentation**, meaning, all pixels with the **same motion** are grouped as one object, **regardless of their semantic category**. This naturally leads to a low evaluation score for our model.
> > > * To illustrate such an effect, we use an example, for an image of a person riding a bike, in single-object segmentation benchmarks, both person and bike are **assigned the same label in ground truth**, as they are undergoing the same motion,  while our prediction will assign **different labels to person and bike.**
> > > * During evaluation on such a benchmark, our model naturally **“over-segment”** the scene from the perspective of motion segmentation, and we can only compute mIoU between ground truth and person, or between ground truth and bike,  in any case, our model gets penalised incorrectly, even though the model has correctly segment the bike and person as two categories.
> > > * For example, for the example provided in Fig 14 in FD,
> > >     * Multi-object evaluation ((a) and (b)): [ IoU(bicycle slot, bicycle mask) + IoU(person slot, person mask) ] / 2
> > >     * Single-object evaluation ((a) and (c)): IoU(person slot, bicycle + person mask)
> > >
> > > (ii) We agree with the reviewer that this is an interesting and important research direction to pursue. We believe that learning pixel correspondence might play an important role by improving predictions around the boundaries. In our early experiments, to provide the model with pixel-level information, we experimented with reconstructing optical flow and extra consistency losses for pixel-level correspondence. However, mostly due to unstable and incorrect flow predictions on complex YTVIS videos, the results did not improve. On the other hand, we believe that recent work on long-term pixel tracking [1] might help achieve sharp boundaries with better correspondences on unconstrained videos. We will explore this option in future work.
> > >
> > > [1] Harley et al., Particle Video Revisited: Tracking Through Occlusions Using Point Trajectories, ECCV22

---

> > > > ### Comment · Reviewer_gzBP · 2023-08-21
> > > >
> > > > Thanks for the author's feedback. The clarification of why the performance drops under the setting of single-object evaluation addressed my concern. I will raise my rating to 5.

---

### Official Review · Reviewer_xWs8 · 2023-07-07

**Soundness:** 3 good
**Presentation:** 3 good
**Contribution:** 2 fair
**Rating:** 5
**Confidence:** 4

**Summary:**

This paper proposes an unsupervised segmentation method in videos. The backbone network is pre-trained with the self-supervised method. Then, the model spatially binds objects to slots on each frame and then relates these slots across frames. The framework is trained to reconstruct the middle frame in a high-level semantic feature space.

**Strengths:**

1. The proposed method is able to segment a real work video without supervision by utilizing a powerful pre-trained feartures.
2. The performance of the proposed method is better than comparison methods.
3. The method is evaluted on both sythethic and real-world dataset.

**Weaknesses:**

1. Although the method can roughly segment the objects in videos, the method is not capable of detecting the boundary of objects.
2. The method can only be applied to videos since it relies on temporal information.

**Questions:**

1. What's the advantage of the proposed model compared to the recent Segment Anything model?
2. In slot merging, whether the clustering method will affect the performance?
3. What if the method is applied to more complex scenes which have more than 12 objects?

**Limitations:**

see weakness

---

> ### Author Rebuttal · Authors · 2023-08-09
>
> We are grateful for the reviewer's feedback and perceptive remarks. We have provided responses and hope these will resolve any issues you've raised.
> ## W1. The method is not capable of detecting the boundary of objects
> We agree with the reviewer. Our method currently only segments objects at patch-level (due to the ViT backbone), and this causes a loss of pixel-level details at the boundaries. However, given more computation resources available, scaling the image resolution, reducing the patch size, or using feature pyramid may resolve this limitation, we leave this as future work.
>
> ## W2. The model relies on temporal information
> We agree with the reviewer. We believe that using temporal information for self-supervised video object segmentation is the right way to go, as we humans are experiencing dynamic scenes everyday, rather than static frames.
>
> ## Q1. Differences to SAM
> The fundamental difference lies in the source of supervision signals. Specifically, SAM is a supervised model, trained with 11M images and 1B+ masks, which is extremely costly to produce, while in contrast, our proposed model is trained in a self-supervised manner, by that we mean, the entire training procedure **does not** require any manual annotations.
>
> ## Q2. Different clustering algorithms
> We cannot use other clustering algorithms that require a predefined number of clusters. In agglomerative clustering, the number of clusters are dynamically set based on a threshold. For more details, please refer to the Q4 of the global response.
>
> ## Q3. Slot number
> Please refer to Q3 of the global response.

---

> > ### Comment · Reviewer_xWs8 · 2023-08-18
> > **Reply to Author Rebuttal**
> >
> > I have read other reviews and the author's rebuttal. Thanks for the response which addressed part of my concerns. I agree with the authors that SAM is a supervised model. However, it generalizes well to unseen scenes. Since we already have this kind of model, what's the point of designing from scratch instead of working based on it? I think a zero-shot segmentation method based on SAM holds greater significance than the proposed method.

---

> > > ### Author Response · Authors · 2023-08-19
> > >
> > > Thank you for your valuable feedback and time.
> > >
> > > (i) While we acknowledge the potential of SAM, we would like to emphasise the importance of object-centric learning without any annotation or extra modalities. From a scientific perspective, we explore the potential of object-centric learning without any labels, pursuing the same ability as human infants can achieve. This, we believe, can spark off novel solutions to the multi-object segmentation challenge. The unsupervised nature of our model allows for training with diverse video data without expensive labour to obtain manual annotations.
> > >
> > > (ii) Following the reviewer’s suggestion, we started exploring the potential of SAM for our task. We tested it on some YTVIS frames and found that SAM consistently over-clusters objects into parts. This requires post-processing techniques to merge parts into objects, for example, similar to our slot merging approach. This is an interesting direction to pursue in future work. Lastly, while we are working on this project, SAM was not released, in fact, it is only accepted by ICCV23, and according to the review policy, we should still treat it as an unpublished paper.

---

### Official Review · Reviewer_Sj2S · 2023-07-26

**Soundness:** 3 good
**Presentation:** 3 good
**Contribution:** 3 good
**Rating:** 5
**Confidence:** 3

**Summary:**

This paper proposes a new self-supervised method for multi-object segmentation in videos called SOLV (Self-supervised Object-centric Learning for Videos). It adopts axial spatial-temporal slot attention to group pixels into slots within frames and then relates these slots across frames to track objects. It uses the masked autoencoder training objective to reconstruct visual features from latent slots. A slot merging strategy based on agglomerative clustering is proposed to address over-segmentation by dynamically merging similar slots. Experiments show state-of-the-art results on the MOVi-E synthetic dataset and Youtube-VIS 2019 real videos without requiring additional modalities like optical flow.

**Strengths:**

The paper is well-written and easy to follow. The problem setup and overall approach are clearly explained.
The performance looks great. The masked feature reconstruction seems interesting.
The spatial-temporal binding and the slot merging make sense to me.

**Weaknesses:**

A comparison with self-supervised baselines on the real data (DAVIS17 / YTVIS19) needs to be added, as there is a huge performance gap between synthetic data and real data (80 vs 30). Such a huge gap makes it questionable whether the results on synthetic data is really representative. Also, it would be great to also add recent supervised methods to help readers understand the gap between self-supervised / supervised methods.

It would be great also to add an error analysis to showcase the failure mode and model behaviors.

It seems that the model needs a superior pretrained backbone (DINOv2) to achieve good performance. DINOv2 is pretrained on a large, curated dataset LVD-142M, which gives the authors an advantage over the baselines. A more fair comparison should be made in Table 2 when compared with DINOSAUR, which is equipped with DINO. I encourage the authors use more text to describe the necessity of using a well-pre-trained encoder instead of simply saying using a frozen encoder in the model.

Given those concerns, I feel like the current results do not show enough evidence for acceptance, I'd be happy to increase my rating if the authors could address my concerns by showing more results (especially for real data with the same visual encoder).

Misc:
I suggest the authors revise Fig.2 and its caption to make it more self-contained. Now the illustration is unclear and has many symbols that are not explained in the caption, making it difficult to understand without referring to the text.

**Questions:**

Please see the weakness part.

---

> ### Author Rebuttal · Authors · 2023-08-09
>
> Thank you for the positive feedback and valuable comments, we have provided detailed clarification, hope these can resolve your concern, thus raise the score accordingly.
>
> ## W1. Comparisons on real data
> We agree with the reviewer that performance on synthetic data is questionable, this is exactly the reason to benchmark on real videos. As far as we know, our work is the first model that tries to segment multiple objects on **real-world data**, without relying on any other modality, for example, optical flow, depth etc.
>
> In Table 2, we have compared our model to the recent SOTA unsupervised video segmentation model, namely, OCLR, that is trained by using optical flow, and SOTA image-based unsupervised segmentation, namely, DINOSAUR (with a powerful DINOv2 backbone), in Table 3.
>
> Supervised models for unsupervised video object segmentation generally exploits pretrained object detectors, as requested by the reviewer, for reference, existing supervised SOTAs of UVOS in DAVIS17 validation set are Propose-Reduce [1] and UnOVOST [2] with mIoUs of 67.0 and 66.4, respectively. We will add these numbers into the table in our final paper.
>
> ## W2. Error analysis
> Thanks for the suggestion. In fact, we have actually provided common failure cases in Supplementary (line 136 and Figure 13), we will move those to the main text of the final paper with more explanations.
>
> Generally speaking the failure mode can be cast into four categories:
>
> **(i) lack of sharp boundaries:** This is because we encode and process the image on the patch-level (14 x 14 patches), therefore cannot recover some of the pixel-level details such as boundaries while upsampling the patch-level segmentation results into the original resolution.
>
> **(ii) overclustering:** This refers to the case where a single object is represented by multiple slots, this often occurs when the object has distinct visual features that are hard to be grouped together.
>
> **(iii) underclustering nearby instances:** When multiple instances of the same semantic class are represented by a single slot, it occurs due to their similar features and positional encodings. This similarity, stemming from their spatial proximity, results in their reconstruction using just one slot.
>
> **(iiii) missing small objects:** When encountering small objects, the model often doesn't allocate a separate slot for them because they minimally impact the reconstruction loss.
>
> Qualitative examples of failure cases can be found in Supplementary, Figure 13.
>
> ## W3. The model needs a superior pre-trained backbone
> Thank you for the comments. In order to compare with DINOSAUR fairly, we have re-trained it with a DINO-v2 backbone, as explained in line 263. Comparing Model-A (DINOSAUR) and Model-E (ours) in Table 3, it can be observed that ours substantially outperform the DINOSAUR model, using the same backbone, i.e., 37.8 vs. 45.3 mIOU. In addition, we also experimented with the DINO backbone in Table 5, and it can be seen, our model even outperforms DINOSAUR with DINO-v2 backbone, i.e.,  37.8  vs. 41.9 mIOU. We have provided more discussion about the effect of different backbones (specifically DINO vs DINOv2) and provide qualitative results in Supplementary. More examples with DINO and DINO-v2 are shown in Fig 10, 11 in FD. In some cases, DINO backbone performs better than DINO-v2, mostly due to more distinct features of different regions leading to overclustering  (Fig 12, 13 in FD).
>
> The DINO series has been widely used for unsupervised object segmentation in the literature. As suggested by the reviewer, we will add DINOSAUR with DINOv2 backbone (model-A) to Table 2 as well, and add more discussion on the necessity of using a well-trained self-supervised backbone in line 293 paragraph.
>
> ## W4. Misc
> Thank you for the suggestion, we will revise the caption of the main figure and explain each symbol and operation briefly.
>
> **References**
>
> [1] Lin et al., Video Instance Segmentation with a Propose-Reduce Paradigm, ICCV21
>
> [2] Luiten et al., Unsupervised Offline Video Object Segmentation and Tracking, WACV20

---

> > ### Comment · Reviewer_Sj2S · 2023-08-18
> >
> > Thanks for the authors' response, the comparison now makes sense to me. I will raise my rating to borderline accept

---

> > > ### Author Response · Authors · 2023-08-19
> > >
> > > Thank you for re-evaluating our work and adjusting the score. We sincerely appreciate your time and thoughtful feedback.

---

### Official Review · Reviewer_xM1B · 2023-07-26

**Soundness:** 2 fair
**Presentation:** 3 good
**Contribution:** 2 fair
**Rating:** 5
**Confidence:** 4

**Summary:**

This paper proposes a Self-supervised Object-centric Learning framework for unsupervised multi-object video segmentation. To achieve this, this paper proposes to derive object-centric representations in a self-supervised manner to facilitate video segmentation tasks. The proposed approach adopts axial spatial-temporal slots attention to group visual regions with similarity properties, while exploiting the training techniques of masked feature reconstruction and slot merging. Experimental results on synthetic and real video datasets demonstrate superior performance against previous works.

**Strengths:**

1. Learning object-centric representations in an unsupervised fashion is vital yet challenging for several downstream applications, such as video object segmentation or robot manipulation. The task aiming to be addressed in this paper is significant.
2. The experiment results and analysis both in the main paper and supplementary material are comprehensive.
3. The overall paper is easy to follow.

**Weaknesses:**

My primary concern lies in the novelty and significance of the proposed framework. The novelty and contributions of the proposed approach are unclearly described in this paper. In the current paper draft, though it employs and modifies a series of existing techniques, it is not evident how the proposed approach is novel or provides significant advancements over existing methods. For example, COMUS [A] also learns self-supervised object-centric representations from DINO, aiming to achieve unsupervised object segmentation. The spatial binding is slightly modified from invariant slot attention [3], and the temporal binding is simply realized by the self-attention mechanism. In the visual decoder, the slot merging process is based on Agglomerative Clustering (AC) algorithm, while the decoder architecture follows DINOSAUR [62]. In summary, the proposed method seems to be a combination of existing approaches to perform the task of multi-object video segmentation, it is currently unclear how their combination in this framework constitutes a significant improvement or novelty over existing methods.

[A] Zadaianchuk et al., Unsupervised Semantic Segmentation with Self-Supervised Object-Centric Representations. ICLR 2023

**Questions:**

1. This proposed method utilizes a pre-defined number of slots to allow multi-object segmentation (e.g., 8 slots). Can the trained model be applicable to an image with more than 8 objects?
2. In Table 3, why did adding spatial and temporal binding (Model-C and D) deteriorate the FG-ARI score compared with Model-B?
3. Also in Table 3, without temporal binding (Model-D) just slightly drops the performance of the full version (Model-E). Does it imply that the temporal information would not be necessarily considered if the frame-by-frame segmentation is well performed?

**Limitations:**

The authors have listed the limitations of this proposed method and the potential future research directions to alleviate these raised issues.

---

> ### Author Rebuttal · Authors · 2023-08-09
>
> We appreciate the reviewer's thorough review and insightful comments. We have offered comprehensive clarifications and hope these will address your concerns.
>
> ## W1. Concern on novelty and significance
> Please refer to Q1 of the global response for our contributions. We would like to highlight that our model is distinct from COMUS [1]. COMUS clusters features of object proposals at the dataset level for image-level semantic segmentation and does not employ any slot-attention.
>
> ## Q1. On the number of slots vs. the number objects
> Please refer to Q3 of the global response.
>
> ## Q2. FG-ARI gets worse while adding spatial-temporal binding
> Please refer to Q2 of the global response.
>
> **References**
>
> [1] Zadaianchuk et al., Unsupervised Semantic Segmentation with Self-Supervised Object-Centric Representations. ICLR23

---

> > ### Comment · Reviewer_xM1B · 2023-08-18
> >
> > Thank you for your response and clarification. I have reviewed the comments from the other reviewers as well as the authors' responses. While the rebuttal for novelty (Q1) does detail the functionality of each module and mentions experimental comparisons and analysis provided in the main paper, it still doesn't clearly highlight the primary novel designs (e.g., architecture, training objectives) for this task. Most of the modules described in the main paper/response seem to be adopted or only slightly modified from existing techniques. As such, my initial concern—that the proposed framework seems to be a combination of existing approaches to perform the task—has not been sufficiently addressed. Therefore, I've chosen to maintain my original score.

---

> > > ### Author Response · Authors · 2023-08-19
> > >
> > > Thank you for taking the time to provide detailed feedback on our submission.
> > >
> > > To emphasise the novelty, our method is **the first unsupervised approach capable of segmenting multiple objects on complex real-world videos**, which is valuable on its own since **no previous work has had any success on the YTVIS dataset before**. While we build on previous work for some of the individual components, we made some specific design choices to make it work and the importance of each design choice is ablated in our experiments. We hope that the reviewer appreciates the effort that goes into adapting, integrating, and experimenting with each of these modules to address **unique and novel challenges on real data.**

---

> > > > ### Comment · Reviewer_xM1B · 2023-08-21
> > > >
> > > > Thank you for the clarifications provided by the authors. The contribution of this work lies in the design of an effective framework for unsupervised multi-object segmentation in real-world videos. Given that, I have raised my rating to 5. As for the novelty concerns, the clarifications provided in the rebuttal are encouraged to be incorporated and highlighted in the revised version.

---

### Author Rebuttal · Authors · 2023-08-09

We appreciate all reviewers for their valuable comments and feedback. We hope the following response can fully resolve the raised concerns. For referred images, please see the **Figure Document (FD)** attached below.

## Q1. Contribution Summary
We would like to start the rebuttal by elaborating our contributions in this paper:

**(i) On the considered task:** We propose the first unsupervised video multi-object segmentation model that can work on both real-world and synthetic videos, without relying on any other modality, for example, optical flow, depth etc.

**(ii) On architecture design:** we propose a conceptually simple architecture that consists of
* **DINO backbone:** *We are the first to address multi-object segmentation in real-world videos by performing reconstruction in the feature space of DINO.* Furthermore, we perform masked feature reconstruction which provides efficiency in time and memory (as pointed out by reviewers Sj2S, gzBP)
* **S-Bind & T-Bind:** *We extract spatially invariant slots (S-Bind) and relate them to each other through time (T-Bind).* Our spatial binding module is a modified version of ISA, as we mentioned in the paper. T-Bind attends slots with the same index across time for consistent slot representations. The two binding modules provide complementary information to slots
* **Slot Merging:** *Given distinctive slot representations learned by our model, we merge slots to solve over-clustering issue.* Over-clustering is a well-known issue in unsupervised object-centric segmentation due to predefined number of slots. Our merging module adopts a simple agglomerative clustering to tackle this problem, as will be detailed in response to Q4
* **Overall Pipeline:** *We propose a distinct architecture compared to previous work*. Specifically, existing object-centric methods for videos follow a similar approach by simply propagating the updated slot information to the next frame through the video. We adopt a totally different approach by extracting slots in frame level, relating them temporally and training it with feature reconstruction

**(iii) Comparison with other approaches:** Compared to other methods, ours outperforms in synthetic data (Table 1) and handles real-world videos. Ablation studies validate our binding modules' efficacy (Table 3) and emphasise the memory efficiency of our masked feature reconstruction (Fig 7). Our slot-merging addresses the over-clustering issue prevalent in object-centric models (Table 4)

## Q2. Behaviours of Models
Reviewers ask about the performance variance between models in our model ablation table (Table 3).
In our ablation study, two key observations emerge:
* FG-ARI, which we report to follow the common practice, is not an accurate indicator of segmentation quality in videos, as mIoU is the widely accepted metric for such tasks
* Both model-C and model-D tend to over cluster objects when compared to model-B

We explain the reasons below with examples in detail:
* FG-ARI considers only in-frame segmentation. FG-ARI does not consider the consistency of indices across frames since it is calculated per frame. On the other hand, mIoU evaluates temporal consistency
* FG-ARI becomes either 1 or 0 when there is only one object. FG-ARI calculation becomes problematic when there is a single object to evaluate. Assume that there is only one labelled object and 4 pixels for it:
`ARI([0, 0, 0, 0], [0, 1, 1, 1]) = 0`. Therefore, if there is only one GT object with a small mistake in the mask, FG-ARI considers it completely wrong. This works for the benefit of model-B roughly segmenting the objects (Fig 1, 2 in FD)
* Each binding module increases the slot specialisation. As reflected in quantitative results, combining the two binding modules increases the total consistency, both for merging and tracking. Slot specialisation is reinforced by both temporal information and invariant visual representations. To show this behaviour qualitatively, we visualise the similarity matrices comparing slot representation inframe and between consecutive frames for model-C, model-D, and model-E in Fig 3, 4, 5 in FD, respectively
* Due to slot specialisation, over-clustering occurs more in model-C and model-D. After inspecting specific cases where model-D and model-C have higher mIoU but lower FG-ARI than model-B, we found that increased slot specialisation leads to overclustering. Specialisation improves tracking, but FG-ARI does not get affected by errors in tracking. Please see Fig 6 in FD to compare segmentations of model-B and model-D for a sample video.
* Overall, due to (i) issues with FG-ARI (ii) tendency to over cluster, model-C and model-D have a lower FG-ARI than model-B. On the other hand, full model (model-E) reaches the FG-ARI of model-B also with a large improvement in mIoU (7.09).

## Q3. Number of Slots vs. Number of Objects
The model tends to undercluster the scene if the predefined number of slots is less than the optimal number. Specifically, our model has a tendency to merge nearby objects into a single slot (Fig 7 (a), (b) in FD) and misses some objects in the background (Fig 7 (c) in FD) when the optimal number of slots is higher than the predefined one.

## Q4. On Slot Merging with Agglomerative Clustering
Empirically, we found that parts of an object might be assigned to separate slots, though these slots still have similar representations (please refer to Fig 3 of the paper). Agglomerative Clustering (AC) works by merging data points based on similarity until a group similarity threshold is reached, therefore, we group similar slots into a dynamically defined number of clusters with AC. It does not need to set a predefined number of clusters as in k-means, and instead, it can dynamically determine the number of groups based on an empirically found similarity threshold. We provide some clustering examples with different threshold values, using the slots of the model trained without slot merging in Fig 8, 9 in FD.

---

### Author Response · Authors · 2023-08-15
**Kind Reminder**

Dear Reviewers,

We would appreciate it if you could share with us whether our rebuttal has addressed your concerns. We would also be happy to answer if you have any further questions.

Thank you

---

### Decision · Program_Chairs · 2023-09-21

**Decision:**

Accept (poster)

**Comment:**

This paper proposes a Self-supervised Object-centric Learning framework for unsupervised multi-object video segmentation. To achieve this, this paper proposes to derive object-centric representations in a self-supervised manner to facilitate video segmentation tasks. The proposed approach adopts axial spatial-temporal slots attention to group visual regions with similarity properties, while exploiting the training techniques of masked feature reconstruction and slot merging. During the review stages, issues like clarification/motivation, novelty/technical contributions, and experiment/justification were raised. The authors also provided detailed responses to address the raised issues, and the reviewers all lean toward the acceptance rating. Therefore, I think this paper is above the threshold for publication.